# InvertNAS: An Invertible Architecture Performance Predictor for Neural Architecture Search

## Abstract

Neural Architecture Search (NAS) aims to find high-performing models, with candidate evaluation often being the most expensive step. While NAS-Bench datasets facilitate the development of performance prediction models by providing benchmark results, most existing work focuses on improving predictor accuracy, with limited attention to search strategies and the selection of initial architectures used for training. In this work, we reformulate NAS as an inverse problem of performance prediction by utilizing Invertible Neural Networks (INNs) to construct a bidirectional performance prediction model that maps architectures to performance and, inversely, maps performance targets back to architectures. Specifically, we train the performance predictor and the search strategy together, in an end-to-end manner. We further propose a novel sampling strategy that selects promising initial architectures without requiring any candidate training. Experiments show that InvertNAS outperforms state-of-the-art NAS methods on NAS-Bench-201, and NAS-Bench-NLP, and performs competitively on NAS-Bench-101 and NAS-Bench-301. These results demonstrate the effectiveness and query efficiency of our approach. We believe this inverse formulation provides a promising direction for future NAS research.

## 1 Introduction

Neural architecture search (NAS) aims to automatically discover neural network architectures that can outperform manually designed ones. A typical NAS algorithm consists of three components (Elsken et al., 2019): the search space, the performance estimation strategy, and the search strategy. The primary goal of a NAS algorithm is to identify multiple promising architectures within the search space. Based on different performance estimation strategies, NAS methods can be categorized into several types: vanilla NAS (Zoph & Le, 2017), which trains each architecture from scratch; one-shot NAS (Bender et al., 2018; Liu et al., 2019), which trains a supernetwork with weight sharing to approximate performance and reduce evaluation time; zero-cost proxy-based NAS (Mellor et al., 2021; Abdelfattah et al., 2021; Yamasaki et al., 2025; Gracheva, 2025), which estimates performance based on architectural features without training; and predictor-based NAS (Wen et al., 2020; Wu et al., 2021; Lukasik et al., 2021; Jing et al., 2022), which employs a surrogate model to predict the performance of neural architectures. Among these, vanilla NAS is the most resource-intensive. Meanwhile, one-shot NAS and zero-cost proxy methods cannot evaluate architectures based on actual performance. Predictor-based NAS leverages a performance predictor to identify promising architectures for training, significantly reducing evaluation time. Although the predictor's accuracy improves with more training architectures, collecting such data is costly, as each architecture must be trained to obtain its performance label. This overhead becomes a limitation when the search space is large. Furthermore, while existing predictor-based NAS methods focus heavily on improving prediction accuracy, relatively little attention has been paid to the design of search strategies or the selection of initial architectures used for predictor training, which are both crucial for effective and efficient search.

We can interpret performance prediction in neural architecture search (NAS) as a regression task, where the goal is to estimate the accuracy of a given architecture based on its structural representation. From this perspective, NAS can be viewed as the inverse problem: instead of predicting the

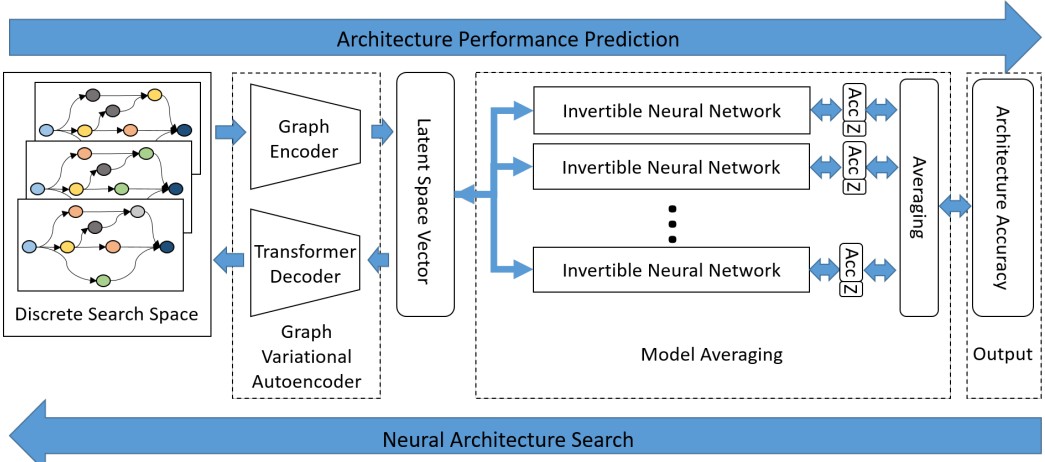

Figure 1: Illustration of the model architecture and workflow. From left to right: the architecture undergoes a graph encoder transformation into a latent space code, which is processed by $N$ INNs. The accuracy values are aggregated using inverse distance weighting to produce the final prediction. From right to left: The target accuracy, paired with a random vector $z$, is inverted through the $N$ INNs parallelly to produce the latent representation, which a transformer decoder then translates into $N$ architectures.

performance from a known architecture, we aim to identify the architecture that yields a desired level of performance. If we could accurately reverse the performance prediction process, then training a performance predictor would be functionally equivalent to solving the NAS problem itself. However, the nonlinearity introduced by activation functions in neural networks makes it difficult to compute the inverse of a standard model. Invertible Neural Networks (INNs) (Ardizzone et al., 2018; 2019) address this issue by ensuring bijectivity, meaning that the mapping from inputs to outputs is reversible. This property allows the output to be mapped back to its corresponding input. To construct such a model, standard dense layers can be replaced with coupling layers commonly used in INNs.

We propose a predictor-based NAS method, InvertNAS, which constructs an invertible performance predictor by combining a graph variational autoencoder (GVAE) (Kipf & Welling, 2016) with an invertible neural network (INN) (Ardizzone et al., 2018). InvertNAS supports bidirectional mapping between neural architectures and their performance (as illustrated in Figure 1), enabling both prediction and inverse generation. This design allows us to train the performance predictor and search strategy jointly in an end-to-end fashion, improving the alignment between prediction and search efficiency. Using unsupervised learning, we train an autoencoder to align neural architectures with their corresponding latent representations. By interpreting the NAS performance predictor as an encoder, we map architectures into a latent space, which is then used to predict their accuracies. The invertible neural network allows us to derive a latent representation by querying a target performance value. This representation is subsequently passed through the GVAE decoder to generate the corresponding architecture $x$. As a result, InvertNAS supports a bidirectional mapping between neural architectures and their performance. Our approach leverages the invertibility of the model to enable a global search over the architecture space. This alternative search strategy offers unique advantages and opens new directions for NAS research. In addition, we introduce a novel sampling strategy for selecting initial architectures to train the performance predictor. Our method identifies informative and promising candidates with minimal overhead, providing a more effective alternative to random selection. Our contributions are summarized as follows:

- Introduce invertible neural networks into NAS for the first time, enabling bidirectional architecture-performance mapping.
- Propose InvertNAS, which jointly trains a performance predictor and search strategy via an GVAE-INN framework.
- Propose a sampling strategy that selects informative initial architectures with minimal overhead, serving as an effective alternative to random selection.

- Demonstrate that InvertNAS outperforms state-of-the-art (SOTA) methods on NAS-Bench-201 and NAS-Bench-NLP, and performs comparably on NAS-Bench-101 and NAS-Bench-301, highlighting its overall effectiveness.

## 2 RELATED WORKS

Neural architecture search (NAS) is typically formulated as a discrete optimization problem over a graph-based search space, where nodes represent operations and edges represent connections. When represented by an operation matrix and an adjacency matrix, this space becomes difficult to optimize using gradients. Early NAS methods employed discrete encodings and adopted strategies such as random search (RS) (Li & Talwalkar, 2020), local search (LS) (Den Ottelander et al., 2021; White et al., 2021b), Bayesian optimization (BO) (Kandasamy et al., 2018), and evolutionary algorithms (EA) (Real et al., 2019). To address the inefficiency of these methods, differentiable NAS was proposed. DARTS (Liu et al., 2019), for instance, relaxed the search space into a continuous form by introducing soft operator selection, enabling gradient-based optimization of both architecture parameters and model weights. The development of tabular NAS benchmarks, such as NAS-Bench-101 (Ying et al., 2019) and NAS-Bench-201 (Dong & Yang, 2020), has further accelerated NAS research by significantly reducing the cost of architecture evaluation. Recent works focus on learning more effective architectural representations. With the help of graph neural networks (GNNs), architectures can be embedded into a continuous latent space, facilitating more efficient search. GA-NAS (Rezaei et al., 2021) leverages reinforcement learning to discover promising patterns, while AG-Net (Lukasik et al., 2022) iteratively refines architectures using a graph-based decoder. Delta-NAS (Sridhar & Chen, 2025) introduces a performance-difference predictor and integrates it into an evolutionary framework to improve sample efficiency. GENNAPE (Mills et al., 2023) and FLAN (Akhauri & Abdelfattah, 2024) improve the encoding quality, although they do not emphasize search strategies. CR-LSO (Rao et al., 2022) embeds the discrete space into a convex latent space to enable gradient-based optimization. Diffusion-based approaches (An et al., 2023; Asthana et al., 2024) have also shown strong performance, with DiNAS (Asthana et al., 2024) achieving competitive results across multiple benchmarks (state of the art). In contrast to prior work, we reformulate NAS as an inverse learning problem, where architecture search and performance prediction are unified within a single framework. Our method leverages invertible neural networks to enable bidirectional mapping and incorporates a sampling strategy that selects informative initial architectures without training candidate architectures, thereby improving both efficiency and search quality.

## 3 METHOD

We interpret the NAS performance predictor as an encoder that maps architectures to accuracy. Under this view, particularly when only limited sampled data is available, architectures can first be projected into a latent space and subsequently mapped to their corresponding accuracies. This formulation enables the use of unsupervised learning to train an autoencoder that aligns neural architectures with latent representations. In this framework, the encoder of the autoencoder maps architectures into the latent space, while the decoder reconstructs architectures from the latent representations. We then introduce an invertible neural network (INN) to model the relationship between the latent space and accuracy. Unlike conventional models that learn a unidirectional mapping from the latent space to performance $y$, the INN allows for bidirectional training: both from latent space to $y$ and from $y$ to latent space. This bidirectional capability enables the INN to capture a richer and more complete relationship between latent representations and accuracy. The capabilities of the invertible function are discussed in Appendix C.

Building on this framework, we propose InvertNAS, which supports both forward prediction and inverse retrieval, enabling direct navigation of the search space toward high-performing architectures. The overall workflow is illustrated in Figure 1. The architecture-to-performance prediction process follows the left-to-right direction in Figure 1. An input architecture is first encoded into a latent representation via a graph encoder. This latent representation is then processed by $N$ independent invertible neural networks, each producing a predicted accuracy. The outputs are aggregated using inverse distance weighting (IDW) (Shepard, 1968) to obtain the final predicted performance. For further implementation details of IDW, please refer to Appendix D. Conversely, the NAS process is represented by the right-to-left direction in Figure 1. A target accuracy is used as the input, paired

with a random vector $z$. This combination is passed through the inverse of the $N$ INNs to obtain the corresponding latent representations. These codes are then decoded by a transformer-based decoder to generate $N$ candidate architectures. The details of each component are described in the following subsections. Finally, we introduce our proposed sampling strategy for selecting initial architectures to train the performance predictor before retraining.

## 3.1 MODEL ARCHITECTURE

**Graph Variational Autoencoder (GVAE)**   Our encoder, which references Arch2Vec (Yan et al., 2020), employs the Graph Isomorphism Network (GIN) layer (Xu et al., 2018) to construct our graph encoder. We utilize a five-layer GIN with hidden sizes of 128, 128, 128, 128, 16. The decoder is a multi-head transformer-based model with positional embedding. It is configured with 3 transformer blocks, 32 embedding dimensions, and 256 hidden dimensions. Therefore, the dimension of latent representations is $|V| \times 16$, where 16 is the GIN output channels and $|V|$ is the number of nodes in a graph $G$. The transformer decodes the latent representation into two parts: operation and adjacency matrix. Both parts are treated as classification tasks. The operation task selects the operation types for each node, while the adjacency matrix task predicts the existence of an edge. Hence, the tensor shape of operations is $(|V|, |OPS|)$, where $|OPS|$ is the number of candidate operations. The tensor shape of the adjacency matrix is $(|V|, |V|, 2)$, viewed as a binary classification task.

The objective of the GVAE is to reconstruct the input graph from its latent representation while simultaneously learning a meaningful latent space. The objective function for this process is formulated as the sum of three components: the node classification cross-entropy loss, edge classification cross-entropy loss, and the Kullback-Leibler (KL) divergence loss. The node classification loss, $\mathcal{L}_{node}$, quantifies the discrepancy between the predicted node labels and the true node labels, encouraging accurate classification of nodes in the reconstructed graph. Similarly, the edge classification loss, $\mathcal{L}_{edge}$, measures the difference between the predicted edge labels and the true edge labels, promoting precise edge classification in the reconstructed graph. In addition to these reconstruction losses, the GVAE incorporates the KL divergence loss, $\mathcal{L}_{KL}$, which encourages the learned latent space to adhere to a prior distribution, typically a multivariate Gaussian distribution. By including this loss term, the GVAE promotes the learning of a smooth and structured latent space, facilitating meaningful interpolation and exploration of graph structures during inference. The overall objective function for the GVAE is defined as:

$$\mathcal{L}_{GVAE} = \alpha_{gvae}\mathcal{L}_{node} + \beta_{gvae}\mathcal{L}_{edge} + \gamma_{gvae}\mathcal{L}_{KL}, \tag{1}$$

where $\alpha_{gvae}$, $\beta_{gvae}$, and $\gamma_{gvae}$ represent the trade-offs for these terms, and we set them to 1.0, 1.0, and 0.16, respectively, across all experiments. By optimizing this objective function, the GVAE aims to reconstruct the input graph accurately while simultaneously learning a latent representation that captures relevant graph properties and enables efficient generation of new graph instances.

**Invertible Neural Network (INN)**   The INN architecture is based on the work presented by (Ardizzone et al., 2018). We made slight modifications to certain hyperparameters, such as the number of layers and hidden sizes. We utilize 4 coupling layers and 5 hidden layers with a hidden size of 256 for our INN. For the aggregated model, we employ multiple INNs and use their mean squared error (MSE) to perform inverse distance weighting.

The training objectives of the INN include the regression loss, reverse loss, and latent loss. Each objective serves a distinct purpose during training. The regression MSE loss, denoted as $\mathcal{L}_{reg}$, is a supervised loss used to assess the model's accuracy in predicting the target output. This loss measures the difference between the predicted output $\hat{y}$ and the ground truth $y$, encouraging the INN to reduce this discrepancy. The reverse MSE loss, denoted as $\mathcal{L}_{rev}$, is an unsupervised term that ensures the invertibility of the INN. It evaluates the difference between the original input $x$ and the reconstructed input $\hat{x}$, which is obtained by applying the inverse operation of the INN to the predicted output $\hat{y}$. Minimizing this loss encourages the INN to reconstruct the input accurately and to establish a bijective mapping between the input and output spaces. The latent Maximum Mean Discrepancy (MMD) loss, denoted as $\mathcal{L}_{latent}$, is also unsupervised. Its objective is to align the latent variable $z$ with a prior distribution and to eliminate the dependency between the predicted output $\hat{y}$ and the random latent variable $z$. The overall objective function for training the INN is a weighted sum of these losses:

$$\mathcal{L}_{INN} = \alpha_{inn}\mathcal{L}_{reg} + \beta_{inn}\mathcal{L}_{rev} + \gamma_{inn}\mathcal{L}_{latent}, \tag{2}$$

where $\alpha_{inn}$, $\beta_{inn}$, and $\gamma_{inn}$ are the weighting coefficients for each loss term. We set these coefficients to 5.0, 1.0, and 10.0, respectively, for all initial training for InvertNAS. Optimizing this objective supports the goals of accurate prediction, invertibility, and disentanglement of the latent space.

During the retraining phase, the loss function in Equation 2 is modified using the rank-based weighting strategy proposed in AG-Net (Lukasik et al., 2022). This strategy emphasizes architectures with higher accuracy across the training set. By applying rank-based weights, data samples with higher accuracy are assigned greater importance, while those with lower accuracy are given less weight. The loss function used during retraining is defined as follows:

$$
\begin{aligned}
\mathcal{L}_{INN} = \frac{1}{\kappa|B| + rank\,(x_i, B)} \cdot (\alpha_{inn}\mathcal{L}_{forward} + \beta_{inn}\mathcal{L}_{rev}) + \gamma_{inn}\mathcal{L}_{latent}, \\
where\ rank\,(x_i, B) = |\{x_j : Acc\,(x_j)\ >\ Acc\,(x_i)\,, x_j \in B\}|\,, \\
for\ x_i \in B, i = 1, \ldots, |B|
\end{aligned}
\tag{3}
$$

## 3.2 Selecting Initial Architectures

Ideally, the initial set of architectures should exhibit a distribution that closely approximates that of high-performing architectures within the search space. However, evaluating all possible architectures is computationally infeasible. As a result, many existing methods simply sample architectures randomly from the search space. An alternative strategy is to use zero-cost proxies to select promising candidates. Although these proxies are more efficient, they often suffer from high estimation errors and limited generalizability. To address these limitation, we leverage AlignFlow (Grover et al., 2020), a model shown to be effective in graph-to-graph translation and domain adaptation. A detailed description of the AlignFlow framework can be found in Appendix E. In our approach, we employ AlignFlow to connect the reduced and original NAS search spaces by learning a shared latent space using normalizing flows. This alignment enables smooth and meaningful translation between the two spaces. Building on this framework, we construct a more effective initial architecture set by first defining a reduced search space that retains the original adjacency matrix but uses a simplified set of operations. Within this subspace, we employ the zero-cost proxy to identify promising candidate architectures. These candidates are then projected back to the full search space via the latent representation learned by AlignFlow, and any invalid architectures are removed. A detailed description of the reduced search space is provided in Appendix F.5. This process allows us to initialize the performance predictor with informative architectures without requiring exhaustive evaluation or training in the full search space.

## 3.3 InvertNAS Searching Algorithm

As described in Algorithm 1, during each iteration of the search process, we first generate candidate architectures by sampling vectors $z$ from a normal distribution with standard deviation $std$. The inverse path of the InvertNAS model is then used to decode these latent vectors into architectures $g$. Any decoded architecture that already exists in the labeled set $A_{visit}$ is discarded, and new candidates are accumulated into the set $G_{cand}$. If no new architecture is found, the standard deviation $std$ is gradually increased to introduce more variation in the latent space and improve exploration. This retry process is repeated until either the number of retries exceeds $R_{max}$ or at least 100 new candidates are collected. Next, the forward path of the model predicts the performance of all candidates in $G_{cand}$. This prediction process does not necessitate the utilization of the query budget. Subsequently, we choose the top-k architectures to query their actual performances. Their true performance is then evaluated, which consumes part of the query budget $b$. These evaluated architectures are added to the labeled set $A_{visit}$. After each iteration, the model is retrained using the updated labeled set for $e$ epochs. The search process continues until the query budget is exhausted. Finally, the architecture in $A_{visit}$ with the highest accuracy is returned.

---

**Algorithm 1:** InvertNAS Search Process

---

**Input:** InvertNAS model $Model$, query budget $b$, retrain epochs $e$, labeled set $A_{visit}$, max retries
$\quad\quad R_{max}$

1 **while** $|A_{visit}| < b$ **do**
2 $\quad$ $G_{cand} \leftarrow \emptyset$, $std \leftarrow 0$, $retry \leftarrow 0$
3 $\quad$ **while** $retry < R_{max}$ **or** $|G_{cand}| < 100$ **do**
4 $\quad\quad$ Sample $z \sim N(0, std)$
5 $\quad\quad$ $g \leftarrow Model.\text{inverse}(1.0, z)$
6 $\quad\quad$ $A_{temp} \leftarrow g - A_{visit}$
7 $\quad\quad$ **if** $|A_{temp}| = 0$ **then**
8 $\quad\quad\quad$ $std \leftarrow std + \tau$
9 $\quad\quad$ $G_{cand} \leftarrow G_{cand} \cup A_{temp}$
10 $\quad\quad$ $retry \leftarrow retry + 1$
11 $\quad$ Select top-$k$ from $Model.\text{predict}(G_{cand})$
12 $\quad$ $A_{visit} \leftarrow A_{visit} \cup \text{eval}(G_{cand})$
13 $\quad$ Retrain $Model$ on $A_{visit}$ for $e$ epochs
14 **return** *Architecture in* $A_{visit}$ *with highest accuracy*

---

## 4 EXPERIMENTS

We evaluate InvertNAS and perform ablation studies on the initial sampling strategy. Our implementation uses an ensemble of 10 INNs, and we compare it with a variant using a single INN to analyze the benefits of ensemble modeling. The implementation is built using TensorFlow (Abadi et al., 2015) and Spektral (Grattarola & Alippi, 2021) for model construction and graph neural network (GNN) operations. We also partially incorporate code from Mehta et al. (2022); Yan et al. (2021); Ardizzone et al. (2018); Gracheva (2025) to facilitate experimental setup and evaluation. We evaluate our approach on two widely used tabular benchmarks, NAS-Bench-101 (NB101) and NAS-Bench-201 (NB201), as well as the surrogate benchmark NAS-Bench-301 (NB301) and NAS-Bench-NLP (NBNLP). The details of the datasets are listed in Appendix F. All experiments are conducted over 10 independent trials, and the mean results are reported. We adopt BANANAS (White et al., 2021a), random search (RS) (Li & Talwalkar, 2020), local search (LS) (White et al., 2021b), and regularized evolution (RE) (Real et al., 2019) as comparison methods. The results of other NAS algorithms are taken directly from their original publications. Methods marked with ‡ indicate results reported in the corresponding papers, while those marked with † refer to results reported in Lukasik et al. (2022). An asterisk (∗) denotes the optimal-performing method. The hyperparameters used in our experiments are listed in Appendix H.

### 4.1 EXPERIMENTAL RESULTS

**NAS-Bench-101** Table 1 presents a comparison of InvertNAS and existing NAS algorithms on the NB101 benchmark. Overall, InvertNAS demonstrates strong and consistent performance, surpassing prior baselines such as BANANASWhite et al. (2021a), RS, LS, and RE in both validation and test accuracy. The single-INN variant, InvertNAS (single), achieves a validation accuracy of 95.01% and a test accuracy of 94.21%, outperforming most existing methods using 192 queries. The full InvertNAS model further improves performance, reaching the optimal validation accuracy of 95.06% and a test accuracy of 94.23%, which is the second highest test accuracy in NB101. Compared to DiNAS, which reports a validation accuracy of 94.98% and a test accuracy of 94.27% with 150 queries, InvertNAS achieves slightly higher validation accuracy and comparable test accuracy, while demonstrating substantially improved stability across runs. Although the test accuracy of InvertNAS is marginally lower, its zero standard deviation indicates significantly higher reliability, compared to the 0.20% reported for DiNAS.

**NAS-Bench-201** Table 2 compares InvertNAS with various baseline and SOTA NAS algorithms on NB201 across CIFAR-10, CIFAR-100, and ImageNet16-120. Overall, InvertNAS demonstrates strong and consistent performance, matching or surpassing leading methods such as BANANAS, AG-

Table 1: Comparison of NAS algorithms on NB101.

| Method | Val. Acc (%) | Val. StD (%) | Test Acc (%) | Test StD (%) | Queries |
|--------|--------------|--------------|--------------|--------------|---------|
| **Optimal** | **95.06** | - | **94.32** | - | - |
| BANANAS[†] | 94.73 | 0.14 | 94.09 | 0.19 | 192 |
| RS[†] | 94.31 | 0.15 | 93.61 | 0.27 | 192 |
| LS[†] | 94.57 | 0.15 | 93.97 | 0.13 | 192 |
| RE[†] | 94.47 | 0.11 | 93.89 | 0.20 | 192 |
| AG-Net[†] | 94.90 | 0.22 | 94.18 | 0.10 | 192 |
| DiNAS[‡] | 94.98 | 0.17 | 94.27 | 0.20 | 150 |
| InvertNAS (single) | 95.01 | 0.09 | 94.21 | 0.04 | 192 |
| InvertNAS | 95.06* | 0.00 | 94.23 | 0.00 | 150 |

Table 2: Comparison of NAS algorithms on NB201.

| Method | CIFAR-10 | | CIFAR-100 | | ImageNet16-120 | | Queries |
|--------|----------|----------|-----------|----------|----------------|----------|---------|
| | Val. Acc | Test Acc | Val. Acc | Test Acc | Val. Acc | Test Acc | |
| **Optimal** | **91.61** | **94.37** | **73.49** | **73.51** | **46.73** | **47.31** | - |
| $\beta$-DARTS[‡] | 91.55 | 94.36 | 73.49* | 73.51* | 46.37 | 46.34 | - |
| BANANAS | 91.55 | 94.26 | 73.49* | 73.51* | 46.68 | 46.49 | 192 |
| RS | 91.27 | 94.02 | 72.12 | 72.31 | 45.67 | 46.08 | 192 |
| LS | 91.53 | 94.31 | 72.28 | 73.25 | 45.44 | 46.77 | 192 |
| RE | 91.48 | 94.94 | 72.86 | 72.98 | 46.04 | 46.43 | 192 |
| AG-Net[†] | 91.60 | 94.37* | 73.49* | 73.51* | 46.64 | 46.43 | 192 |
| Shaply-NAS[‡] | 91.61* | 94.37* | 73.49* | 73.51* | 46.57 | 46.85 | 200 |
| DiNAS[‡] | 91.61* | 94.37* | 73.49* | 73.51* | 46.66 | 45.41 | 192 |
| InvertNAS (single) | 91.61* | 94.37* | 73.49* | 73.51* | 46.51 | 46.84 | 192 |
| InvertNAS | 91.61* | 94.37* | 73.49* | 73.51* | 46.70 | 47.18 | 192 |

Net, Shaply-NAS (Xiao et al., 2022), and DiNAS. The single-INN variant of InvertNAS successfully identifies the globally optimal architectures on CIFAR-10 and CIFAR-100. On ImageNet16-120, which is considered a more challenging dataset, InvertNAS achieves better performance than BANANAS, DiNAS, and Shaply-NAS. The full InvertNAS model obtains the highest validation and test accuracy among all compared methods, reaching 46.70% and 47.18% respectively. These results are achieved using only 192 queries, demonstrating the effectiveness and query efficiency of our method. Moreover, InvertNAS is able to discover the optimal architectures on CIFAR-10 and CIFAR-100 with as few as 150 queries.

**NAS-Bench-301 and NAS-Bench-NLP**  Table 3 presents the performance of InvertNAS on NB301 and NBNLP. Compared with baseline methods such as BANANAS, RS, BO (Kandasamy et al., 2018), and RE, as well as advanced methods including AG-Net and DiNAS, InvertNAS achieves the highest validation accuracy. For NB301, both the single-INN and full-model variants reach 94.94%. Although InvertNAS uses 150 queries, compared to 100 queries in DiNAS, its performance demonstrates that it can attain SOTA accuracy with a modest increase in query cost. On the NBNLP, InvertNAS sets a new SOTA result, reaching 96.21% accuracy, which is 0.15% higher than DiNAS. Additionally, our method has the lowest standard deviation among all compared methods.

**Summary of Experimental Results**  Across NB101, NB201, NB301, and NBNLP, InvertNAS consistently demonstrates SOTA performance with competitive query efficiency. It matches or surpasses leading methods such as DiNAS and AG-Net in terms of validation and test accuracy, while also exhibiting strong robustness with low or zero performance variance. InvertNAS performs reliably across diverse search spaces and datasets, consistently identifying high-performing architectures

Table 3: Comparison of NAS algorithms on NB301 and NBNLP

| Method | NB301 | | | NBNLP | | |
|---|---|---|---|---|---|---|
| | Val. Acc (%) | Std (%) | Queries | Val. Acc (%) | Std (%) | Queries |
| BANANAS[†] | 94.77 | 0.10 | 192 | 95.68 | 0.16 | 304 |
| RS[†] | 94.31 | 0.12 | 192 | 95.64 | 0.19 | 304 |
| BO[†] | 94.71 | 0.10 | 192 | - | - | - |
| RE[†] | 94.75 | 0.11 | 192 | 95.66 | 0.21 | 304 |
| AG-Net[†] | 94.79 | 0.12 | 192 | 95.86 | 0.18 | 304 |
| DiNAS[‡] | 94.92 | 0.07 | 100 | 96.06 | 0.17 | 304 |
| InvertNAS (single) | 94.94 | 0.02 | 150 | - | - | - |
| InvertNAS | 94.94 | 0.01 | 150 | 96.21 | 0.10 | 304 |

under moderate query budgets. These results affirm the practical utility of InvertNAS as a general and effective solution for neural architecture search.

## 4.2 ABLATION STUDIES ON INITIAL SAMPLING STRATEGY

To evaluate the effectiveness of our proposed initial sampling strategy, we compare it against two alternative baselines by replacing our method in the initial selection phase. The experiments are conducted on three NAS benchmarks: NB101, NB201, NB301 and NBNLP, and the results are summarized in Table 4. The first baseline applies a zero-cost proxy, to rank candidate architectures within the target search space, selecting the top-ranked ones. For NB101 and NB201, where the search spaces are relatively small, the proxy is applied to all architectures. For NB301, we randomly sampled 100,000 architectures due to its significantly larger search space and then ranked them using a proxy. For NBNLP, we sampled 200,000 architectures instead. The second baseline adopts uniform random sampling without any proxy guidance. To assess the quality of the selected architectures, we compare their distribution against that of top-performing architectures in the search space. Specifically, we define an ideal distribution by selecting the top 1%, 3%, and 5% of architectures ranked by validation accuracy. We then apply kernel density estimation (KDE) to estimate the rank distributions of the sampled architectures and calculate the Kullback–Leibler (KL) divergence between each sampling strategy and the ideal distribution. Lower KL values indicate a closer alignment with the distribution of high-performing architectures. As shown in Table 4, our proposed method, denoted as AlignFlow, consistently achieves the lowest KL divergence across all settings. This indicates that the architectures selected by AlignFlow are more closely aligned with the distribution of optimal architectures. In contrast, the zero-cost proxy and random selection strategies exhibit larger KL divergence values, reflecting greater deviation from the ideal distribution. These results suggest that our proposed strategy is more effective in identifying informative initial architectures, which likely benefits subsequent performance prediction. While the zero-cost proxy method offers a fast estimation, it is susceptible to noise, which can degrade the quality of the selected samples. The random sampling method performs the worst, as expected, due to the lack of any search guidance. Overall, the observed KL divergence values provide strong empirical support for the effectiveness of our sampling strategy in approximating the distribution of optimal regions in the search space.

We conducted additional ablation studies on hyperparameters, including the number of INNs and different ensemble methods. The results are presented in Appendix J and Appendix K, respectively.

## 5 CONCLUSION

We propose InvertNAS, a novel predictor-based NAS framework that reformulates neural architecture search as an inverse problem of performance prediction. By integrating invertible neural networks with a graph-based autoencoder, our method enables a bidirectional mapping between architectures and their predicted performance. This design allows for end-to-end joint training of the performance predictor and search strategy, improving the overall alignment between architecture evaluation and

Table 4: KL with different sampling strategies with InvertNAS.

| NB101 CIFAR-10 | Val. Acc (%) | Test Acc (%) | KL (1%) | KL (3%) | KL (5%) |
|---|---|---|---|---|---|
| alignflow | 95.06 | 94.23 | 0.000107 | 0.000074 | 0.000062 |
| zc | 95.06 | 94.23 | 0.001460 | 0.002388 | 0.001392 |
| random | 94.87 | 94.17 | 0.003040 | 0.001984 | 0.001916 |
| NB201 CIFAR-10 | Val. Acc (%) | Test Acc (%) | KL (1%) | KL (3%) | KL (5%) |
| alignflow | 91.61 | 94.37 | 0.000480 | 0.000704 | 0.000312 |
| zc | 91.61 | 94.37 | 0.001438 | 0.001654 | 0.001339 |
| random | 91.55 | 94.32 | 0.085604 | 0.052934 | 0.055364 |
| NB201 CIFAR-100 | Val. Acc (%) | Test Acc (%) | KL (1%) | KL (3%) | KL (5%) |
| alignflow | 73.49 | 73.51 | 0.001796 | 0.000876 | 0.000704 |
| zc | 73.49 | 73.51 | 0.007344 | 0.006179 | 0.005385 |
| random | 73.49 | 73.51 | 0.023032 | 0.021633 | 0.065222 |
| NB201 imagenet16 | Val. Acc (%) | Test Acc (%) | KL (1%) | KL (3%) | KL (5%) |
| alignflow | 46.70 | 47.18 | 0.002247 | 0.001740 | 0.000844 |
| zc | 46.68 | 47.08 | 0.005997 | 0.005260 | 0.019844 |
| random | 46.65 | 47.13 | 0.025670 | 0.036503 | 0.068861 |
| NB301 CIFAR-10 | Val. Acc (%) | Test Acc (%) | KL (1%) | KL (3%) | KL (5%) |
| alignflow | 94.94 | - | 0.000034 | 0.000022 | 0.000020 |
| zc | 94.80 | - | 0.000053 | 0.000031 | 0.000023 |
| random | 94.73 | - | 0.000762 | 0.000488 | 0.000357 |
| NBNLP PTB | Val. Acc (%) | Test Acc (%) | KL (1%) | KL (3%) | KL (5%) |
| alignflow | 96.21 | - | 0.000038 | 0.000019 | 0.000014 |
| zc | 96.15 | - | 0.000038 | 0.000019 | 0.000014 |
| random | 96.11 | - | 0.000123 | 0.000087 | 0.000076 |

exploration. We further introduce an efficient sampling strategy that selects informative initial architectures without requiring candidate training. Experimental results on NB101, NB201, NB301 and NBNLP demonstrate that InvertNAS achieves competitive or superior performance compared to SOTA methods, while maintaining high query efficiency and robustness. Especially, InvertNAS achieved a new SOTA on NBNLP. These findings highlight the potential of invertible models for enabling globally informed architecture generation and suggest a promising direction for future NAS research. In particular, extending this inverse formulation to broader search spaces, multi-objective settings, or hardware-constrained environments may further enhance the generalizability and impact of the proposed approach.

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

## A  LIMITATIONS AND BROADER IMPACTS

### A.1  LIMITATIONS

We did not perform hyperparameter tuning for InvertNAS due to computational constraints. Given the unsupervised nature of GVAE and AlignFlow, each component was trained once and reused as a pretrained model in our experiments. While various implementations of invertible neural networks exist, we adopt a representative variant to demonstrate the applicability of INNs to the NAS problem. Our evaluation focuses on NAS-Bench-101, NAS-Bench-201, NAS-Bench-301, and NAS-Bench-NLP. Although additional NAS constraints such as inference time and parameter count were not considered in our current experiments, the invertible structure of INNs allows the predictor to be

naturally extended to incorporate such constraints. This can be achieved by jointly predicting relevant metrics, such as inference time and parameter size, and by padding the latent vector with a random variable $z$ at designated positions. Extending InvertNAS in this direction is left for future work.

## A.2 BROADER IMPACTS

We propose InvertNAS, which, as discussed in the contributions of the introduction, redefines the NAS problem from a new perspective. InvertNAS formulates NAS as the inverse problem of performance prediction, providing a novel theoretical framework that enables the joint consideration of the search strategy and performance estimation strategy. This formulation offers a promising new direction for future NAS research, both in theory and methodology.

Our work may also inspire future studies to extend the concept of invertible neural networks (INNs) to broader areas in automated machine learning (AutoML) and generative design problems. If InvertNAS can identify high-performing architectures with only a small number of queries, it will advance NAS towards a low-cost, high-efficiency paradigm. This not only lowers the barrier to entry, making NAS more accessible to resource-constrained research environments, but also contributes to reducing resource consumption and carbon emissions.

## B LICENSE FOR DATASETS AND CODES

We provide a summary of the licenses for all datasets and codebases used in our work in Table 5. The invertible neural network (INN) layer was adapted from the implementation available at `https://github.com/jaekookang/invertible_neural_networks`. At the time of writing, the license information for both the INN, epsinas codebase and AlignFlow was not publicly disclosed.

Table 5: License information for assets

| Asset | License |
|---|---|
| NB101 | Apache License 2.0 |
| NB201 | MIT License |
| NB301 | Apache License 2.0 |
| nas-bench-x11 | Apache License 2.0 |
| NASLib | Apache License 2.0 |
| Alignflow | Not Found |
| INN | Not Found |
| epsinas | Not Found |

## C IMPLEMENTATION AND EQUATIONS OF INN LAYERS

Invertible Neural Networks (INNs) (Ardizzone et al., 2018; 2019) are designed to establish a reversible mapping between inputs and outputs, enabling both forward and inverse transformations. The model architecture ensures bijectivity, a one-to-one correspondence between inputs and outputs, which is critical for applications requiring inference in both directions. This is achieved by constructing the network using a series of invertible transformations, commonly known as normalizing flows (Tabak et al., 2010; Tabak & Turner, 2013; Trippe & Turner, 2018), which transform a simple base distribution, typically a standard Gaussian, into a complex target distribution. The core building block of an INN is the coupling layer, which operates by splitting the input vector $x$ into two equally sized parts: $x = [x_1, x_2]$. The output is then computed by applying a sequence of learned transformations to each sub-vector. These transformations are defined through the following forward and backward equations:

$$y_1 = x_1 \odot \exp(s_2(x_2)) + t_2(x_2), \quad y_2 = x_2 \odot \exp(s_1(y_1)) + t_1(y_1), \tag{4}$$

$$x_2 = (y_2 - t_1(y_1)) \odot \exp(-s_1(y_1)), \quad x_1 = (y_1 - t_2(x_2)) \odot \exp(-s_2(x_2)). \tag{5}$$

Here, $x_1, x_2, y_1, y_2$ are vectors of equal dimension, and $\odot$ denotes element-wise multiplication. The transformation functions $s_1(), s_2(), t_1(), t_2()$ can be arbitrary neural networks and do not need to be invertible themselves.

During the forward pass, the input $x$ is split into $x_1$ and $x_2$ to compute the outputs $y_1$ and $y_2$ via Equation 4. The final output vector is then formed as $y = [y_1, y_2]$. Conversely, during the inverse pass, $y$ is decomposed back into $y_1$ and $y_2$, and the original input $x = [x_1, x_2]$ is recovered using Equation 5.

Because the forward and inverse computations require the input and output vectors to have identical dimensions, a latent vector $z$ sampled from a standard normal distribution is often concatenated to the output: $y' = [y, z]$. This augmentation preserves dimensional consistency and allows the model to encode information that may be lost in the transformation. The variable $z$ can also be interpreted as capturing uncertainty or stochastic variations in the mapping. The forward and inverse mappings within the $i$-th INN layer can be written more explicitly as:

$$
\begin{aligned}
y_i &= [y_{i1},\ y_{i2}] \\
&= f(x_i) \\
&= f([x_{i1}, x_{i2}]) \\
&= [x_{i1} \odot e^{s_{i2}(x_{i2})} + t_{i2}(x_{i2}), \\
&\qquad x_{i2} \odot e^{s_{i1}(x_{i1} \odot e^{s_{i2}(x_{i2})} + t_{i2}(x_{i2}))} + t_{i1}(x_{i1} \odot e^{s_{i2}(x_{i2})} + t_{i2}(x_{i2}))]
\end{aligned}
\tag{6}
$$

$$
\begin{aligned}
x_i &= [x_{i1},\ x_{i2}] \\
&= f^{-1}(y_i) \\
&= f^{-1}([y_{i1},\ y_{i2}]) \\
&= [(y_{i1} - t_{i2}((y_{i2} - t_{i1}(y_{i1})) \odot e^{-s_{i1}(y_{i1})})) \odot e^{-s_{i2}((y_{i2} - t_{i1}(y_{i1})) \odot e^{-s_{i1}(y_{i1})})}, \\
&\qquad (y_{i2} - t_{i1}(y_{i1})) \odot e^{-s_{i1}(y_{i1})}]
\end{aligned}
\tag{7}
$$

where $x_{i1}, x_{i2}$ and $y_{i1}, y_{i2}$ denote the partitioned components at the $i$-th layer. The same transformation functions $s_{i1}(), s_{i2}(), t_{i1}(), t_{i2}()$ are shared between the forward and inverse passes, ensuring consistency across both directions of the mapping. By stacking multiple INN layers and optimizing them using suitable loss functions—such as regression loss, inverse consistency loss, and latent regularization—the model can effectively learn bidirectional mappings. This makes INNs particularly suitable for tasks that require both generation and inversion capabilities. To the best of our knowledge, this is the first work that applies INNs to the neural architecture search problem.

## D  INVERSE DISTANCE WEIGHT (IDW)

The inverse distance weighting (IDW) method (Shepard, 1968) improves ensemble prediction by assigning higher weights to more accurate predictors. Predictors with outputs closer to the target receive greater weights, while those with larger errors are downweighted. We implement IDW using a basic weighting function, where the mean squared error (MSE) serves as the distance metric. The final prediction $\hat{y}$ is computed as:

$$
\hat{y} = \frac{\sum_{i=1}^{N} w_i y_i}{\sum_{i=1}^{N} w_i}, \quad \text{where } w_i = \frac{1}{\mathrm{MSE}_i + \varepsilon}
\tag{8}
$$

Here, $\mathrm{MSE}_i$ denotes the mean squared error between the $i$-th predictor's output $y_i$ and the ground truth, and $\varepsilon$ is a small constant (set to $10^{-8}$) for numerical stability.

## E  DETAILS OF ALIGNFLOW FRAMEWORK

AlignFlow (Grover et al., 2020) is a generative adversarial network (GAN) framework designed to utilize data from multiple domains for tasks such as cross-domain translation and unsupervised

domain adaptation. Its core idea is to model each domain using a separate normalizing flow, which allows for flexible and expressive density estimation. A central feature of AlignFlow is the ability to learn a shared latent representation $Z$ across domains. This shared space enables exact inference and consistent alignment between domain-specific distributions. To perform cross-domain translation, AlignFlow defines invertible mappings between each domain and the latent space. For example, given domains $A$ and $B$, the mappings $G_{Z \to A}$, $G_{Z \to B}$, and their inverses are constructed such that the transformation from domain $A$ to domain $B$ can be expressed as a composition: $G_{A \to B} = G_{Z \to B} \circ G_{A \to Z}$.

AlignFlow guarantees marginal consistency between domains under a wide range of training objectives. Its invertible structure also ensures exact cycle consistency, meaning that a sample translated from a source domain to a target domain and then back will perfectly reconstruct the original input. During inference, a source sample is projected into the latent space and decoded into the target domain. The shared latent space supports both conditional and unconditional generation, and also allows for smooth interpolation between domains. In our work, we leverage AlignFlow to transfer from a reduced architecture search space to a more complex target space. This facilitates efficient initialization of candidate architectures in neural architecture search (NAS), leading to improved diversity and sampling quality.

# F    Detailed Search Space Descriptions

## F.1    NAS-Bench-101

NAS-Bench-101 is a public dataset that holds performance details of 423,624 different convolutional neural network (CNN) structures. It is created by fully training and testing these structures on the CIFAR-10 image classification task. The goal of NAS-Bench-101 is to give researchers a common platform to fairly compare different NAS algorithms and understand what they're good and bad at. It uses a method called operation-on-nodes. The space of a cell includes all possible operations on $V$ nodes, with the first and last nodes marked as operation INPUT and OUTPUT. They are the input and output tensors to the cell. Each node carries one of $L$ labels. The amount of labeled Directed Acyclic Graphs (DAGs) will increase quickly with both $V$ and $L$. To keep the size of the space down, some limits are set: there are three kinds of operations (1x1 convolution, 3x3 convolution, 3x3 max-pool), the maximum number of nodes is seven, and the most edges allowed is nine. The

## F.2    NAS-Bench-201

NAS-Bench-201 constitutes a dataset incorporating 15,625 potential configurations of neural cells. These architectures have undergone training and evaluation across three distinct image classification tasks, including CIFAR-10, CIFAR-100, and ImageNet-16-120. Drawing parallels with NAS-Bench-101, it constructs its search space employing cells. However, the distinctive feature of NAS-Bench-201 lies in its operation-on-edges mechanism, diverging from the approach in NAS-Bench-101. Each cell is depicted as a DAG composed of four nodes and six edges. The predefined set encompasses five operations, including 1x1 and 3x3 convolutions, 3x3 average pooling, zero operation, and skip connections. Additionally, it provides comprehensive data such as training duration and pertinent metadata for each epoch of every architecture, thereby serving as a valuable resource for researchers in the field of NAS. In the context of our experiments, we have introduced modifications to transform the DAG from an operation-on-edges format into an operation-on-nodes format, thereby ensuring that the data format is consistent with NAS-Bench-101.

## F.3    NAS-Bench-301

The DARTS search space comprises over $10^{18}$ architectures, making tabular benchmarks like NAS-Bench-101 impractical. NAS-Bench-301 addresses this by providing a surrogate benchmark for NAS on CIFAR-10, using a model trained on 60k sampled architectures and their validation accuracies. Each architecture consists of eight stacked cells. The $k$-th cell takes as input the outputs of cells $k-2$ and $k-1$, with $1 \times 1$ convolutions applied as needed for dimension matching. The space defines two cell types: normal and reduction. Reduction cells, placed at one-third and two-thirds of the network depth, use stride-two operations to reduce spatial resolution. All normal (or reduction) cells share a

common architecture. Each cell contains seven nodes. The first two receive inputs from preceding cells; the remaining four are intermediate nodes. Outputs from intermediate nodes are concatenated along the depth dimension. Edges between nodes are assigned one of seven operations: separable conv $3 \times 3$, separable conv $5 \times 5$, dilated separable conv $3 \times 3$, dilated separable conv $5 \times 5$, max pool $3 \times 3$, avg pool $3 \times 3$, and skip connection.

### F.4 NAS-BENCH-NLP

The search space for NAS-Bench-NLP focuses on recurrent neural network architectures. The macro-level architecture is the same as AWD-LSTM (Merity et al., 2018), and the cell structure is defined as follows: the operations for each node include linear transformations, element-wise blending/product/sum, and activation functions such as tanh, Sigmoid, and LeakyReLU. The maximum number of nodes is 24, with a maximum of 3 hidden states and a maximum of 3 linear input vectors. The NAS-Bench-NLP model is trained on the Penn Tree Bank (PTB) dataset. There are a total of $10^5 3$ architectures in the defined search space. Since the original NAS-Bench-NLP dataset is a tabular dataset containing only a subset of the search space defined in the NAS-Bench-NLP paper, we opted to use the surrogate version of NAS-Bench-NLP from NAS-Bench-x11 (Yan et al., 2021). While the official metric for NAS-Bench-NLP is perplexity, we will use the term accuracy for convenience.

### F.5 REDUCED SEARCH SPACE

In our proposed initial sampling strategy for training the InvertNAS, we define the reduced search space for NB101, NB201, and NB301. For NB101 and NB201, we removed the 3x3 convolution operation. For NB301, we removed the separable 3x3 and 5x5 convolution operations, while for NBNLP, the element-wise sum operation was removed. We use NWOT (Mellor et al., 2021) as the zero shot proxy for NB101, NB201 and NB301. We use epsinas (Gracheva, 2025) as the zero shot proxy for NBNLP.

## G DETAILED EVALUATION RESULTS FOR INVERTNAS

Table 6 presents the evaluation results of our proposed InvertNAS method. The variant labeled as single refers to InvertNAS using only a single invertible neural network (INN), while InvertNAS denotes the use of an ensemble of ten INNs. Each result is reported as the mean and standard deviation over ten independent runs. The columns Val. Acc and Test Acc represent the accuracy on the validation and test sets, respectively, while Std denotes the corresponding standard deviation. Compared to the single-INN version, InvertNAS with multiple INNs exhibits a clear reduction in standard deviation, indicating improved stability. Additionally, there is a slight improvement in the average accuracy across most benchmarks.

Table 6: Performance evaluations of InvertNAS

| Dataset | Method | Val. Acc | Val. Std | Test Acc | Test Std | Queries |
|---|---|---|---|---|---|---|
| NB101 | single | 0.9501 | 0.0916 | 0.9421 | 0.0466 | 192 |
| NB201 Cifar10 | single | 0.9161 | 0.0000 | 0.9437 | 0.0000 | 150 |
| NB201 Cifar100 | single | 0.7349 | 0.0000 | 0.7351 | 0.0000 | 150 |
| NB201 ImageNet16 | single | 0.4651 | 0.0012 | 0.4684 | 0.0019 | 192 |
| NB301 | single | 0.9494 | 0.0002 | - | - | 150 |
| NB101 | InvertNAS | 0.9506 | 0.0000 | 0.9423 | 0.0000 | 150 |
| NB201 Cifar10 | InvertNAS | 0.9161 | 0.0000 | 0.9437 | 0.0000 | 150 |
| NB201 Cifar100 | InvertNAS | 0.7349 | 0.0000 | 0.7351 | 0.0000 | 150 |
| NB201 ImageNet16 | InvertNAS | 0.4670 | 0.0006 | 0.4718 | 0.0028 | 192 |
| NB301 | InvertNAS | 0.9494 | 0.0001 | - | - | 150 |
| NBNLP | InvertNAS | 0.9621 | 0.0010 | - | - | 304 |

Following is the comparison of NAS Algorithms on NB201 with Accuracy and Standard Deviation.

Table 7: Comparison of NAS algorithms on NB201 (CIFAR-10) with accuracy and standard deviation.

| Method | Val. Acc (%) | Val. Std (%) | Test Acc (%) | Test Std (%) | Queries |
|---|---|---|---|---|---|
| **Optimal** | **91.61** | **0.00** | **94.37** | **0.00** | - |
| $\beta$-DARTS[†] | 91.55 | 0.00 | 94.36 | 0.00 | - |
| BANANAS | 91.55 | 0.15 | 94.26 | 0.22 | 192 |
| RS | 91.27 | 0.23 | 94.02 | 0.21 | 192 |
| LS | 91.53 | 0.15 | 94.31 | 0.15 | 192 |
| RE | 91.48 | 0.13 | 94.94 | 0.21 | 192 |
| AG-Net[†] | 91.60 | 0.02 | 94.37* | 0.00 | 192 |
| Shaply-NAS[†] | 91.61* | 0.00 | 94.37* | 0.00 | 200 |
| DiNAS[†] | 91.61* | 0.00 | 94.37* | 0.18 | 192 |
| InvertNAS (single) | 91.61* | 0.00 | 94.37* | 0.00 | 192 |
| InvertNAS | 91.61* | 0.00 | 94.37* | 0.00 | 192 |

Table 8: Comparison of NAS algorithms on NB201 (CIFAR-100) with accuracy and standard deviation.

| Method | Val. Acc (%) | Val. Std (%) | Test Acc (%) | Test Std (%) | Queries |
|---|---|---|---|---|---|
| **Optimal** | **73.49** | **0.00** | **73.51** | **0.00** | - |
| $\beta$-DARTS[†] | 73.49* | 0.00 | 73.51* | 0.00 | - |
| BANANAS | 73.49* | 0.00 | 73.51* | 0.00 | 192 |
| RS | 72.12 | 0.90 | 72.31 | 0.92 | 192 |
| LS | 72.28 | 0.52 | 73.25 | 0.58 | 192 |
| RE | 72.86 | 0.83 | 72.98 | 0.79 | 192 |
| AG-Net[†] | 73.49* | 0.00 | 73.51* | 0.00 | 192 |
| Shaply-NAS[†] | 73.49* | 0.00 | 73.51* | 0.00 | 200 |
| DiNAS[†] | 73.49* | 0.00 | 73.51* | 0.00 | 192 |
| InvertNAS (single) | 73.49* | 0.00 | 73.51* | 0.00 | 192 |
| InvertNAS | 73.49* | 0.00 | 73.51* | 0.00 | 192 |

Table 9: Comparison of NAS algorithms on NB201 (ImageNet16-120) with accuracy and standard deviation.

| Method | Val. Acc (%) | Val. Std (%) | Test Acc (%) | Test Std (%) | Queries |
|---|---|---|---|---|---|
| **Optimal** | **46.73** | **0.00** | **47.31** | **0.00** | - |
| $\beta$-DARTS[†] | 46.37 | 0.00 | 46.34 | 0.00 | - |
| BANANAS | 46.68 | 0.09 | 46.49 | 0.42 | 192 |
| RS | 45.67 | 0.52 | 46.08 | 0.60 | 192 |
| LS | 45.44 | 0.18 | 46.77 | 0.25 | 192 |
| RE | 46.04 | 0.54 | 46.43 | 0.38 | 192 |
| AG-Net[†] | 46.64 | 0.12 | 46.43 | 0.34 | 192 |
| Shaply-NAS[†] | 46.57 | 0.08 | 46.85 | 0.12 | 200 |
| DiNAS[†] | 46.66 | 0.09 | 45.41 | 0.59 | 192 |
| InvertNAS (single) | 46.51 | 0.12 | 46.84 | 0.19 | 192 |
| InvertNAS | 46.70 | 0.06 | 47.18 | 0.28 | 192 |

# H IMPLEMENTATION DETAILS

All experiments were conducted on a workstation equipped with 251 GB of RAM, an Intel Xeon Gold 5218 CPU (64 cores), and a single NVIDIA RTX 4090 GPU with 24 GB of memory. We adopt a three-phase training procedure involving GVAE pretraining, AlignFlow-based sampling, and INN optimization.

In the first phase, we pretrain the GVAE component of InvertNAS using self-supervised learning to improve the decoder's ability to reconstruct architectures from latent representations. For NAS-Bench-101 and NAS-Bench-201, the entire search space is used. In contrast, NAS-Bench-301 and NAS-Bench-NLP use a subset of architectures, relying on 50,000 and 25,000 randomly sampled architectures, respectively. The dataset is split into 80% training, 10% validation, and 10% testing. GVAE is trained for 500 epochs with a batch size of 64 (32 for NAS-Bench-301), using the Adam optimizer with a learning rate of $1 \times 10^{-3}$. The noise scale in GVAE is set to 0.05 throughout all experiments. We employ the `ReduceLROnPlateau` scheduler with a factor of 0.1, patience of 50, and minimum learning rate of $1 \times 10^{-5}$, along with early stopping (patience of 100 for NAS-Bench-101 and NAS-Bench-201, and 50 for NAS-Bench-301). On average, training GVAE takes approximately 2 hours.

The second phase involves training AlignFlow to select informative initial architectures. Given the unsupervised nature of GVAE and AlignFlow, each component is trained once and reused throughout all experiments. AlignFlow training takes less than 20 minutes across all benchmarks. We provide the pretrained weights for both GVAE and AlignFlow. In this phase, the AlignFlow generator is trained using Adam with a learning rate of 0.001, $\beta_1 = 0.9$, $\beta_2 = 0.999$, and four coupling layers with hidden dimension 128. The discriminator shares the same learning rate but uses $\beta_1 = 0.5$ and three hidden layers. A validation loss combining generator and discriminator losses is used for early stopping, with patience of 10 and a minimum of 20 epochs. Unlabeled datasets are randomly split into 80% training and 20% validation, with a maximum training size of 5,000 for NAS-Bench-101, 15,625 for NAS-Bench-201, 20,000 for NAS-Bench-301, and 25,000 for NAS-Bench-NLP. We use a batch size of 32 across all datasets.

In the final phase, we freeze the pretrained GVAE and proceed to train the INNs to facilitate architecture search. This training and search process is computationally efficient, typically requiring less than 1.5 GPU hours for a query budget of 192. On average, training the INNs takes approximately 1 hour for NAS-Bench-101 and NAS-Bench-201, and about 1.5 hours for NAS-Bench-301. We begin by initializing the training set with 30 architectures and retain the top 20 based on their performance. In each iteration, we sample 100 candidate architectures and select the top 5 according to predicted performance. The INN model is composed of 10 networks, each with four coupling layers, a latent dimension of 16, five hidden layers with 256 units each, and two coupling blocks. For NAS-Bench-101 and NAS-Bench-201, we use a batch size of 64 and set the learning rate to 0.001 with a `CosineDecay` scheduler, repeating each architecture label 20 times to enhance training stability. For NAS-Bench-301 and NAS-BENCH-NLP, we reduce the batch size to 32. The learning rate is initially set to 0.001 with early stopping (patience of 30) and subsequently decreased to 0.0001 with the same early stopping criterion. Additionally, the label repetition count is reduced to 5 to accommodate the increased complexity of the search space.

Hyperparameter details for all components are summarized in Table 10, Table 11, Table 12, Table 13, and Table 14. The input dimension of the INN is determined by the product of the GVAE latent space dimension and the node encoding dimension within a search cell, which varies across search spaces. For NAS-Bench-101, the node encoding dimension is 7; for NAS-Bench-201, it is 8. In NAS-Bench-301, both the normal and reduction cells use a node encoding dimension of 11, resulting in a combined total of 22. In NAS-Bench-NLP, the node encoding dimension is 20.

Table 10: Hyperparameters for GVAE

| Benchmark | Hyperparameter | Value |
|---|---|---|
| **NB101** | Training epochs | 500 |
| | Patience | 100 |
| | Batch size | 64 |
| | Learning rate | $1 \times 10^{-3}$ |
| | latent space dimension | 16 |
| **NB201** | Training epochs | 500 |
| | Patience | 100 |
| | Batch size | 64 |
| | Learning rate | $1 \times 10^{-3}$ |
| | latent space dimension | 16 |
| **NB301** | Training epochs | 500 |
| | Patience | 50 |
| | Batch size | 32 |
| | Learning rate | $1 \times 10^{-3}$ |
| | latent space dimension | 16 |
| **NBNLP** | Training epochs | 500 |
| | Patience | 50 |
| | Batch size | 32 |
| | Learning rate | $1 \times 10^{-3}$ |
| | latent space dimension | 20 |

Table 11: Hyperparameters for AlignFlow

| Component | Hyperparameter | Value |
|---|---|---|
| **Generator** | Optimizer | Adam |
| | Learning rate | 0.001 |
| | $\beta_1$ / $\beta_2$ | 0.9 / 0.999 |
| | Coupling layers | 4 |
| | Hidden dimension | 128 |
| **Discriminator** | Optimizer | Adam |
| | Learning rate | 0.001 |
| | $\beta_1$ / $\beta_2$ | 0.5 / 0.999 |
| | Hidden dimension | 128 |
| | Hidden layers | 3 |
| **Training** | Validation loss | $generator\ loss + discriminator\ loss$ |
| | Minimum epochs | 20 |
| | Early stopping patience | 10 |
| | Train/Validation split | 8:2 |
| | Batch size | 32 |
| | Training set size (NB101) | 5,000 |
| | Training set size (NB201) | 15,625 |
| | Training set size (NB301) | 20,000 |
| | Training set size (NBNLP) | 25,000 |

Table 12: Hyperparameters for INNs on NAS-Bench-101

| Category | Hyperparameter | Value |
|---|---|---|
| **Shared** | Number of INNs | 10 |
| | Initial training set | Select 30 (top 20 by accuracy) |
| | Validation set size | 10 |
| | Samples per round | 100 |
| | Top-k selection | 5 |
| | Latent dimension | 16 |
| | Coupling layers | 4 |
| | Hidden layers | 5 |
| | Hidden dimension | 256 |
| | Number of couples | 2 |
| **NB101-specific** | Batch size | 64 |
| | Learning rate | 0.001 |
| | Repeat label | 20 |
| | x dimension | 112 |
| | y dimension | 1 |
| | z dimension | 111 |

Table 13: Hyperparameters for INNs on NAS-Bench-201

| Category | Hyperparameter | Value |
|---|---|---|
| **Shared** | Number of INNs | 10 |
| | Initial training set | Select 30 (top 20 by accuracy) |
| | Validation set size | 10 |
| | Samples per round | 100 |
| | Top-k selection | 5 |
| | Latent dimension | 16 |
| | Coupling layers | 4 |
| | Hidden layers | 5 |
| | Hidden dimension | 256 |
| | Number of couples | 2 |
| **NB201-specific** | Batch size | 64 |
| | Learning rate | 0.001 |
| | Repeat label | 20 |
| | x dimension | 128 |
| | y dimension | 1 |
| | z dimension | 127 |

Table 14: Hyperparameters for INNs on NAS-Bench-301

| Category | Hyperparameter | Value |
|---|---|---|
| **Shared** | Number of INNs | 10 |
| | Initial training set | Select 30 (top 20 by accuracy) |
| | Validation set size | 10 |
| | Samples per round | 100 |
| | Top-k selection | 5 |
| | Latent dimension | 16 |
| | Coupling layers | 4 |
| | Hidden layers | 5 |
| | Hidden dimension | 256 |
| | Number of couples | 2 |
| **NB301-specific** | Batch size | 32 |
| | Learning rate | $0.001 \rightarrow 0.0001$ |
| | Repeat label | 5 |
| | x dimension | 352 |
| | y dimension | 1 |
| | z dimension | 351 |

Table 15: Hyperparameters for INNs on NAS-Bench-NLP

| Category | Hyperparameter | Value |
|---|---|---|
| **Shared** | Number of INNs | 10 |
| | Initial training set | Select 30 (top 30 by accuracy) |
| | Validation set size | 10 |
| | Samples per round | 100 |
| | Top-k selection | 5 |
| | Latent dimension | 16 |
| | Coupling layers | 4 |
| | Hidden layers | 5 |
| | Hidden dimension | 256 |
| | Number of couples | 2 |
| **NBNLP-specific** | Batch size | 32 |
| | Learning rate | $0.001 \rightarrow 0.0001$ |
| | Repeat label | 5 |
| | x dimension | 280 |
| | y dimension | 1 |
| | z dimension | 279 |

## I  EVALUATE THE QUALITY OF ARCHITECTURES GENERATED BY THE INVERSE PATH

To evaluate the quality of architectures generated by the inverse path of InvertNAS, we collect statistics on the architectures obtained through InvertNAS's inverse path at each iteration. We then computed three key metrics:

- Validity: The percentage of generated architectures that are valid.
- Uniqueness: The percentage of generated valid architectures that are distinct (not previously encountered).
- Novelty: The percentage of generated architectures that are both valid and entirely new to the search process.

The following Table 16, presents the average of these metrics across all rounds of a single search process for various benchmarks. (Cifar-10, Cifar-100, and imagenet16-120 are benchmarks within NB201.)

As observed from the table, the Validity is consistently close to 100% across most benchmarks, except for imagenet16-120. The lower validity in imagenet16-120 stems from our algorithm's strategy: in earlier stages, when it samples in regions with less noise, validity is near 100%. However, as the algorithm struggles to find new architectures, it increases the sampling noise, leading to a decrease in validity later in the process (approaching 0). At this point, after reaching a retry limit, our algorithm ceases sampling from the inverse path and instead selects the top-k predicted architectures from. This adaptive behavior explains the large standard deviation in validity for imagenet16-120.

Furthermore, the results demonstrate that Novelty and Uniqueness maintain similar levels across all benchmarks. This indicates that the architectures generated through the inverse path in each round are largely consistent and belong to a similar distribution, reflecting the stability of INN.

Table 16: Quality of architectures generated by InvertNAS

| Benchmarks | Validity | Novelty | Uniqueness |
|---|---|---|---|
| Cifar-10 | 1.0000E+00 | 8.0463E-03 | 9.6782E-03 |
| Cifar-100 | 9.9995E-01 | 7.1727E-03 | 8.8784E-03 |
| imagenet16-120 | 3.3101E-01 | 3.0907E-03 | 4.0664E-03 |
| NB101 | 9.9358E-01 | 9.0165E-03 | 1.1164E-02 |
| NB301 | 1.0000E+00 | 1.5388E-02 | 1.4262E-02 |

## J  COMPARISON OF DIFFERENT ENSEMBLE METHODS

We conducted a comparison of different ensemble methods. Our proposed InvertNAS uses Inverse Distance Weighting (IDW) to ensemble the results from the INNs. We compared this with a simple averaged weight approach, where each of the 10 INNs was given an equal weight of 0.1. The results averaged over 10 runs are shown in Table 17. We found that the performance of the Inverse Distance Weighting (IDW) method and the averaged weight method was quite similar across most benchmarks. However, a few notable differences emerged. For both the NB301 validation accuracy and the NB201 ImageNet-16-120 test accuracy, the IDW method yielded slightly better results with a lower standard deviation. This indicates that while both methods are effective, IDW provides a small but consistent improvement in both accuracy and stability.

## K  COMPARISON OF DIFFERENT NUMBERS OF INNS

We conducted an ablation study on the number of Inverse Neural Networks (INNs). Table 18 shows the mean and standard deviation over 10 runs. For simpler benchmarks, such as CIFAR-10 and CIFAR-100 on NB201, we found that varying the number of INNs did not significantly affect the final results, as all configurations were able to find the optimal architecture. However, a greater difference was observed on the ImageNet-16-120 benchmark, which had a larger standard deviation.

Table 17: Comparison of different ensamble methods

|  | Val. Acc | Val. Std | Test Acc | Test Std | Queries |
|---|---|---|---|---|---|
| **Inverse Distance Weight** | | | | | |
| NB101 InvertNAS | 95.06 | 0.00 | 94.23 | 0.00 | 150 |
| NB201 Cifar10 InvertNAS | 91.61 | 0.00 | 94.37 | 0.00 | 150 |
| NB201 Cifar100 InvertNAS | 73.49 | 0.00 | 73.51 | 0.00 | 150 |
| NB201 ImageNet16 InvertNAS | 46.70 | 0.06 | 47.18 | 0.28 | 192 |
| NB301 InvertNAS | 94.94 | 0.01 | - | - | 150 |
| **Averaged Weight** | | | | | |
| nb101 | 95.06 | 0.00 | 94.23 | 0.00 | 150 |
| nb201 cifar10 | 91.61 | 0.00 | 94.37 | 0.00 | 150 |
| nb201 cifar100 | 73.49 | 0.00 | 73.51 | 0.00 | 150 |
| nb201 imagenet | 46.70 | 0.08 | 46.78 | 0.29 | 192 |
| nb301 | 94.92 | 0.01 | - | - | 150 |

These results suggest that for more complex tasks, the number of INNs plays a more significant role in finding the best architecture. Overall, our experiments indicate that a larger number of INNs generally leads to better final performance, as demonstrated by higher accuracy and lower standard deviation across all benchmarks.

Table 18: Comparison of different numbers of INNs

| Benchmark | INN number | Val. Acc (mean, std) | Test Acc (mean, std) |
|---|---|---|---|
| nb301 | 1 | (94.94, 0.02) | - |
|  | 3 | (94.91, 0.04) | - |
|  | 5 | (94.88, 0.03) | - |
|  | 7 | (94.91, 0.00) | - |
|  | 10 | (94.94, 0.01) | - |
| nb101 | 1 | (95.01, 0.09) | (94.21, 0.04) |
|  | 3 | (94.99, 0.09) | (94.21, 0.03) |
|  | 5 | (95.00, 0.12) | (94.23, 0.02) |
|  | 7 | (94.98, 0.11) | (94.19, 0.06) |
|  | 10 | (95.06, 0.00) | (94.23, 0.00) |
| cifar10 | 1 | (91.61, 0.00) | (94.37, 0.00) |
|  | 3 | (91.61, 0.00) | (94.37, 0.00) |
|  | 5 | (91.61, 0.00) | (94.37, 0.00) |
|  | 7 | (91.61, 0.00) | (94.37, 0.00) |
|  | 10 | (91.61, 0.00) | (94.37, 0.00) |
| cifar100 | 1 | (73.49, 0.00) | (73.51, 0.00) |
|  | 3 | (73.49, 0.00) | (73.51, 0.00) |
|  | 5 | (73.49, 0.00) | (73.51, 0.00) |
|  | 7 | (73.49, 0.00) | (73.51, 0.00) |
|  | 10 | (73.49, 0.00) | (73.51, 0.00) |
| imagenet | 1 | (46.51, 0.12) | (46.84, 0.19) |
|  | 3 | (46.65, 0.09) | (46.97, 0.31) |
|  | 5 | (46.73, 0.00) | (46.60, 0.07) |
|  | 7 | (46.70, 0.08) | (46.75, 0.32) |
|  | 10 | (46.70, 0.06) | (47.18, 0.28) |

## L  THE USE OF LARGE LANGUAGE MODELS (LLMS)

We only used the LLMs (ChatGPT and Gemini) to polish the English writing.

