# OpenReview forum: "InvertNAS: An Invertible Architecture Performance Predictor for Neural Architecture Search"
_ICLR.cc/2026/Conference — ICLR 2026 Conference Withdrawn Submission_

### Official Review · Reviewer_FNRY · 2025-10-24

**Soundness:** 2
**Presentation:** 2
**Contribution:** 2
**Rating:** 2
**Confidence:** 5

**Summary:**

This paper introduces InvertNAS, a neural architecture search (NAS) framework that reformulates architecture search as the inverse problem of performance prediction. By integrating INNs with a graph variational autoencoder, the method enables bidirectional mapping between architectures and their performance. Experiments on multiple NAS benchmarks demonstrate that InvertNAS achieves state-of-the-art or highly competitive performance with strong query efficiency.

**Strengths:**

+ This paper contains extensive experiments on NAS benchmarks.

+ The idea of the paper is clearly stated and easy to understand.

**Weaknesses:**

+ The experimental validation of this work is quite limited. Specifically, the experiments are limited on existing NAS benchmarks, such as NAS-Bench-101 and NAS-Bench-201. It is unknown that how the proposed method performs on large-scale datasets such as ImageNet-1K. As a result, the scalability of the proposed method is limited.

+ Because the performance predictor serves to improve the efficiency of NAS methods. Therefore, it is also necessary to report the search cost on large-scale datasets such as ImageNet-1K.

+ The motivation is not really convincing to me. Specifically, the performance predictor only needs to predict the performance of neural architectures thus accelerate the search process of NAS. The reverse process does not seem to play a critical role in achieving this goal.

**Questions:**

+ How does InvertNAS ensure the validity and uniqueness of architectures generated through the inverse path, especially when validity drops significantly in more complex benchmarks like ImageNet16-120?

+ Why does this work not perform hyperparameter study for InvertNAS, and how might this omission affect the reproducibility and fairness of comparisons with state-of-the-art methods?

---

### Official Review · Reviewer_ZFG5 · 2025-10-24

**Soundness:** 2
**Presentation:** 2
**Contribution:** 1
**Rating:** 2
**Confidence:** 5

**Summary:**

This paper presents InvertNAS, a novel NAS method which leverages invertible neural networks to create a bidirectional mapping between architectural representations and their performance. This approach allows not only for predicting the accuracy of a given architecture but also for generating high-performing architectures from a target accuracy. Combined with an efficient sampling strategy for initial architecture selection, InvertNAS achieves state-of-the-art or highly competitive results across multiple NAS benchmarks.

**Strengths:**

+ The framework of the proposed InvertNAS method is clearly presented and easy to understand.

+ The experiments on existing benchmarks are relatively comprehensive.

**Weaknesses:**

+ The bidirectional capability of InvertNAS, while theoretically interesting, offers little practical benefit over standard forward-only predictors, as performance prediction in NAS typically only requires accurate forward estimation.

+ The evaluation is confined to tabular benchmarks (e.g., NAS-Bench-101/201/301, NBNLP) and lacks validation on large-scale real-world datasets like ImageNet-1K, raising concerns about its applicability beyond synthetic settings.

+ It remains uncertain whether the invertible framework can be effectively integrated with other types of NAS algorithms (e.g., differentiable or reinforcement learning-based methods), limiting its perceived generalizability.

+ The use of multiple INNs and complex components like AlignFlow seems to introduce significant computational and architectural complexity, and there lacks a comprehensive time complexity analysis.

**Questions:**

+ How does the AlignFlow-based sampling strategy select initial architectures more effectively than zero-cost proxies and random sampling?

+ In what way does the bidirectional mapping of INNs improve both performance prediction and architecture generation compared to unidirectional predictors?

---

### Official Review · Reviewer_ofpN · 2025-11-01

**Soundness:** 2
**Presentation:** 2
**Contribution:** 2
**Rating:** 2
**Confidence:** 4

**Summary:**

This paper proposes InvertNAS, a novel predictor-based method for Neural Architecture Search. The core contribution is the reformulation of performance prediction as an invertible, bidirectional problem. The authors utilize a Graph Variational Autoencoder (GVAE) to map architectures into a latent space, and then employ an Invertible Neural Network (INN) to create a bidirectional mapping between this latent space and the architecture's performance.

**Strengths:**

1. The primary strength of this work is its novel conceptualization of predictor-based NAS. Using an Invertible Neural Network to unify the performance predictor and the search strategy is an elegant idea. This framework is, to my knowledge, new to the field. The latent z vector theoretically allows the model to sample a diverse distribution of architectures that all map to the same high-performance target.

2. The paper does not just focus on the predictor but also addresses the "cold-start" problem by proposing an initial architecture sampling strategy. The use of AlignFlow to map a proxy-ranked, reduced search space back to the full search space is a clever, non-trivial approach.

**Weaknesses:**

1. The paper's central claim of "outperforming state-of-the-art" (in abstract and conclusion) is a significant overstatement. On the most saturated benchmarks, NAS-Bench-201 (CIFAR-10 and CIFAR-100), the method matches SOTA by finding the known optimal architecture, which several other methods (AG-Net, DINAS, etc.) also accomplish (Tables 2, 7, 8). This is not an outperformance.

2. Insufficient Comparison to Zero-Cost (ZC) Proxies: The experimental comparison omits a critical and highly relevant baseline: NAS methods based entirely on zero-cost (ZC) proxies. The paper dismisses ZC proxies in the introduction as having "high estimation errors," but this overlooks their proven power in ranking architectures. ZC-proxy-based search methods are a dominant SOTA competitor and are orders of magnitude more efficient, requiring no expensive training or querying. The paper only compares against a ZC proxy for initialization (Table 4), not as a full search method, which is a major gap in the evaluation.

3. The proposed method is a highly complex, multi-stage pipeline (GVAE pretraining, AlignFlow training, and an ensemble of 10 INNs. This pipeline introduces a large number of non-trivial design choices and hyperparameters. These crucial parameters need ablation studies to justify.

**Questions:**

1. Clarification of "ZC" in Table 4: In Appendix F.5, the paper states it is NWOT for most and 'epsinas' for NBNLP. This crucial detail should be in the main paper, not buried in an appendix. Why are these proxies selected for comparison?

2. The loss functions for the GVAE (Eq 1) and INN (Eq 2) are weighted sums of multiple components. How were these specific weighting coefficients (e.g., 1, 1, 0.16 for GVAE; 5, 1, 10 for INN) determined? They appear arbitrary and are not justified or ablated, making it difficult to assess their impact on the model's success.

3. Algorithm 1 is confusing.
Line 6: The notation A_{temp}←g−A_{visit} is unconventional for what seems to be a simple check for uniqueness.
Line 11 vs. 12: Line 11 states "Select top-k from Model.predict(G_{cand})," which is consistent with the text ("we choose the top-k architectures to query"). However, Line 12 A_visit is updated with G_cand, seeming to imply that there is no selection.
​
4. Runtime vs. Query Cost: The paper focuses on the number of queries but omits the wall-clock runtime from the main experimental tables. While Appendix H mentions the search process takes "less than 1.5 GPU hours for a query budget of 192," this is a key efficiency metric and should be stated clearly in the main paper alongside the query-based results for a fair comparison. What about query costs in GPU hours for other methods?

---

### Official Review · Reviewer_6zJX · 2025-11-03

**Soundness:** 2
**Presentation:** 2
**Contribution:** 2
**Rating:** 2
**Confidence:** 5

**Summary:**

This paper introduces InvertNAS, a new predictor-based Neural Architecture Search (NAS) framework that reformulates NAS as the inverse problem of performance prediction. The method combines a Graph VAE with Invertible Neural Networks (INNs) to create a bidirectional mapping between architectures and their predicted performance. The model can then both predict the performance of a given architecture, as well as generate architectures corresponding to a desired performance level. The authors also propose a sampling strategy to select informative initial architectures without explicit training. Experiments on common NAS benchmarks such as NAS-Bench-101, -201, -301, and NAS-Bench-NLP show that InvertNAS performs competitively to standard NAS methods.

**Strengths:**

1. Reformulating NAS as an inverse problem via invertible neural networks is an interesting idea and could be a meaningful contribution to performance-predictor NAS research. The end-to-end joint training of a predictor and search module is appealing and more coherent than many existing two-stage NAS methods.

2. The reported results are strong across four standard NAS benchmarks, with relatively small query budgets.

3. The authors provide their code in the supplementary material.

**Weaknesses:**

While the idea of treating NAS as an inverse prediction problem is appealing, this work has several weaknesses which I list below:

1. The practical advantage of INNs compared to other performance predictors and generative models remains unclear. Why did the authors decide about such modelling choice? A more careful review of related work should be provided by the authors, including discussion and comparison to works such as [DiffusionNAG](https://proceedings.iclr.cc/paper_files/paper/2024/file/171c3678c36e39fc0074f3e7332a9a66-Paper-Conference.pdf).

2. The methodological novelty is limited. The method relies heavily on preexisting components from previous work, such as GVAE (Arch2Vec), INN, AlignFlow, in a relatively ad hoc way. The overall architecture seems more like a stack of known modules than a unified methodological innovation.

3. All experiments are on standard NAS benchmarks. It is unclear whether the approach scales to real-world large search spaces or multi-objective tasks (e.g., accuracy vs. latency). See [HW-GPT-Bench](https://arxiv.org/pdf/2405.10299) and the recent [Jet-Nemotron](https://arxiv.org/pdf/2508.15884) as examples of more practical benchmarks. Furthermore, even on the NAS benchmarks used in the submission, InvertNAS is only slightly better than existing methods and some recent baselines are missing too (e.g. DiffusionNAG). Given the system’s added complexity, it is unclear whether these small improvements justify the proposed formulation.

**Questions:**

1. How do you ensure the inverse mapping generates valid architectures? How often are invalid or duplicate architectures produced in practice?

2. Could other generative models (e.g., diffusion-based NAS or even LLMs) achieve similar results?

---

### Note · Authors · 2025-11-18

**Comment:**

We sincerely thank the reviewers and the Area Chair for the time and effort dedicated to reviewing our submission. The constructive feedback provided is greatly valued, and we are committed to improving the quality of our work for future submissions. We apologize for any inconvenience that our withdrawal may cause.

**Withdrawal Confirmation:**

I have read and agree with the venue's withdrawal policy on behalf of myself and my co-authors.